# Automatic reagent handling and assay processing of human biospecimens inside a transportation container for a medical disaster response against radiation

Adam R. Akkad[1], Jian Gu[1]*, Brett Duane[1], Alan Norquist[1], David J. Brenner[2], Adarsh Ramakumar[3], Frederic Zenhausern[1,4]*

**1** Center for Applied NanoBioscience and Medicine, Department of Basic Medical Sciences, The University of Arizona College of Medicine, Phoenix, AZ, United States of America, **2** Center for Radiological Research, Columbia University, Vagelos College of Physicians and Surgeons, New York, NY, United States of America, **3** Armed Forces Radiobiology Research Institute, Uniformed Services University of the Health Sciences, Bethesda, MD, United States of America, **4** HonorHealth Research and Innovation Institute, Scottsdale, AZ, United States of America

* jgu10@arizona.edu (JG); fzenhaus@arizona.edu (FZ)

**Data Availability Statement:** All relevant data are within the paper and its Supporting Information files.

## Abstract

Biological materials can be shipped off-site for diagnostic, therapeutic and research purposes. They usually are kept in certain environments for their final application during transportation. However, active reagent handling during transportation from a collection site to a laboratory or biorepository has not been reported yet. In this paper, we show the application of a micro-controlled centrifugal microfluidic system inside a shipping container that can add reagent to an actively cultured human blood sample during transportation to ensure a rapid biodosimetry of cytokinesis-block micronucleus (CBMN) assay. The newly demonstrated concept could have a significant impact on rapid biodosimetry triage for medical countermeasure in a radiological disaster. It also opens a new capability in accelerated sample processing during transportation for biomedical and healthcare applications.

## Introduction

Biological materials are often shipped and processed off the collection/manufacturing site for diagnostic, therapeutic and research purposes. The quality of the materials must be maintained during transportation for their final laboratory applications. For example, a cold chain (dry ice) or other novel approach was used for shipping cells [1]; an RNA stabilizer (e.g. PAXgene Blood RNA Kit) was used to stabilize the RNA in blood for gene expression assay [2]; a cytokine was used to activate fresh natural killer cells to treat malignancy [3]; and 37°C shipping incubator was used for transporting retinal organoids or primate embryos [4, 5]. In all these cases, samples were treated before being packaged, and desired temperatures were maintained during transportation; however, actively processing the sample for accelerating its analysis during transportation has not been reported. Here we show active addition of reagents to

**Funding:** The funding was awarded to J.G. and F.Z. by a subaward (Agreement No.: 3042) from Armed Forces Radiobiology Research Institute through a Congressionally Directed Medical Research Program, Merit Based Discovery Award to A.R. (Award# W81XWH-15-2-0076, Grant PR142006). The funders (other than the named authors) had no role in study design, data collection and analysis, decision to publish, or preparation of the manuscript.

human blood cell samples in culture during transportation for running of the standard cytokinesis-block micronucleus (CBMN) assay for a smart and more efficient biodosimetry.

## CBMN logistics for rapid biodosimetry triage

Mass nuclear catastrophe is a serious concern for society at large when considering the rising threat of terrorism and the risks associated with harnessing nuclear energy [6]. It can also be a great concern to military personnel due to potential nuclear threats during their operations [7]. In the case of a mass nuclear/radiological event that requires hundreds of thousands of individuals to be assessed for radiation dose exposure, a rapid biodosimetry triage tool is crucial [6, 7]. An ideal biodosimetry triage tool should allow quick sampling and rapid assay/result as soon as possible at high throughput, with the samples and the measurement endpoints remaining valid over a clinically reasonable period to assess dose [8]. Currently no single biodosimetry method can meet all the needs of nuclear triage [7, 8]. It is expected that an array of dosimetry tools will be applied as needed, such as lymphocyte depletion kinetics (LDK), electron paramagnetic resonance (EPR), cytogenetic dicentric (DCA) or CBMN assays, gamma-H2AX assay etc.

To illustrate the capability of different dosimetry approaches, Flood et al. conducted a comparison of several dosimetry methods for both military and civilian triage applications [8]. The dosimetry response time has been shown to be determined by multiple parameters, including availability of supplies, time to collect samples, conditions of sample transportation to laboratory, assay processing throughput, and time to communicate the results back to the individuals [8]. Among different biodosimetry assays, CBMN is a promising assay to address the challenge of rapidly triaging large amounts of people exposed to ionizing radiation. For the CBMN assay, samples can be collected immediately from a blood draw to assess DNA damage [6, 9], and the endpoint measurement can be stable for a year [10]. The dose range covers the critical exposure threshold of 2 Gy, above which immediate medical attention is warranted [11]; recently, it was also extended to 10 Gy [12] that could be useful to triage at 6 Gy, above which delayed treatment could be adopted in a scarce resources-crisis so that those "most likely to survive" can be treated first [11]. To address the bottleneck of sample collection, a self-administered fingerstick blood collector was demonstrated for CBMN assay [2]. The assay was also automated using commercial high-throughput liquid handling robotics [13] that are easy to scale up, and the cell culture time was shortened to 54 hours versus the traditional 72 hours [14].

Despite all the development of CBMN assay for radiation triage, there is still a factor that could significantly delay the response time of CBMN assay, i.e. sample transportation from a disaster site to a laboratory. Due to the complexity of the assay, CBMN samples are usually shipped to a cytogenetic laboratory for high-throughput analysis. However, the shipping time can be significant (especially for civilian applications) and could be as much as 3 days considering different factors such as weather, aircraft mechanical issues etc. To address this issue, we have previously reported a novel logistic solution for pre-conditioning biospecimen for CBMN assay using a 37°C temperature-controlled shipping container built upon a traditional shipping box [15]. The shipping container allows the cell culture of CBMN samples immediately after collection and during transportation, which can dramatically shorten the assay response time from ~ 6 days to ~ 3 days.

However, running a complete CBMN assay does require the addition of the cytokinesis inhibitor cytochalasin B (cyt-B) precisely within a 24–44 hour window (preferably at 24 hours) following the start of cell culture to ensure only first division cells are captured [9]. Fig 1 shows the timelines of the CBMN logistics with or without cell culture during transportation, as well

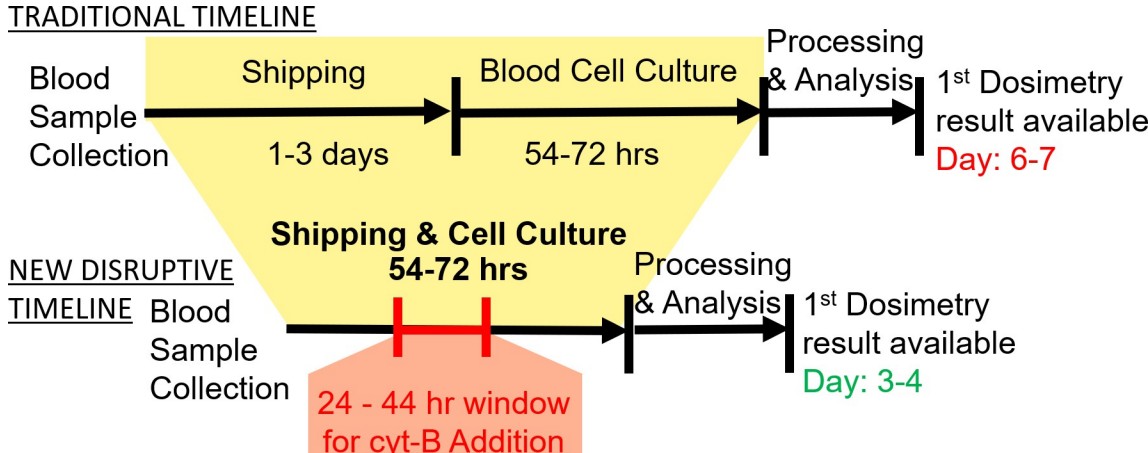

**Fig 1. Timelines of the traditional and new disruptive CBMN assay logistics.** Addition of cyto-B within a 24–44 hour window duration transportation is critical for the success of the new biodosimetry logistics to dramatically cut down the response time, which will be implemented by our SSI.

as the time window for adding cyt-B. It shows that cell culture during transportation can dramatically cut down the response time of CBMN assay, but if the transportation time run over 44 hours, it could cause inaccurate dose results due to the delayed addition of cyt-B.

To address this issue, we upgraded function of our shipping container to include means for the active and controlled addition of cyt-B to the cultured cell samples during transportation by implementing a new configuration with a centrifugal microfluidic system, namely a "Smart Shipping Incubator" (SSI). The SSI includes a custom centrifugal system inside a commercial shipping incubator, and a micropipette tip to release cyto-B during the shipment while also resisting routine shipping mechanical shocks. The design of the SSI system will be described below. Cyto-B release from the micropipette and its resistance to mechanical shocks will also be characterized. The effect of HEPES buffer on the yield of binucleated cells will be reported. We will also show that the CBMN assay results using SSI during transportation correlate well with those using a traditional $CO_2$ laboratory incubator, showing SSI's potential for biodosimetry disaster response and medical emergency applications.

## Results

### SSI system

The SSI system is comprised of three main components. The first is a commercial custom-made iQ5 shipping incubator from Micro Q Technologies, Scottsdale, AZ (Fig 2A). The incubator has outer dimensions of 53.3cm x 53.3cm x 53.3cm (L x W x H) and a weight of 20.4 kg. It is equipped with a rechargeable nickel-metal hydride (NiMH) battery pack that can thermo-electrically heat the unit to maintain an internal temperature of 37˚C. The internal chamber of the incubator has a cylindrical shape with an inner diameter of 24.5 cm and height of 17.5 cm. The chamber wall was made of Aluminum for good thermal conduction.

The second component of the SSI is a custom-made centrifugal system. Fig 2B shows the schematics of the system. The system consists of three assemblies, the top Motor Assembly (MA), the middle Spin Disk Assembly (SDA) and the bottom Bearing Assembly (BA). The BA is mounted to the bottom of the incubator chamber with a hexagonal recess structure at the center of the bearing. A matching hexagonal structure at the bottom of the SDA can be inserted into the recess so that the SDA can spin freely through the bearing. The SDA has four

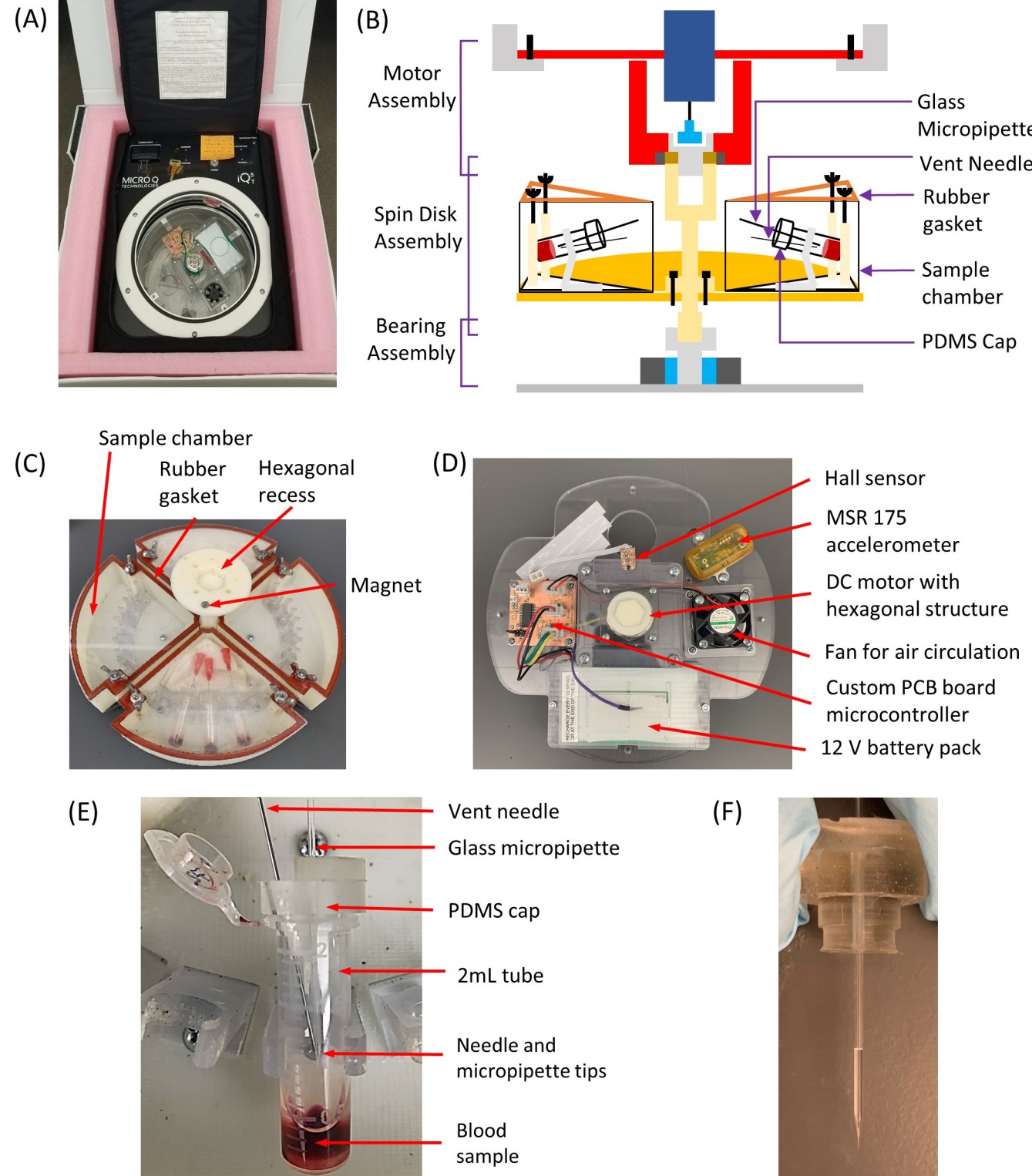

**Fig 2. The Smart Shipping Incubator (SSI).** (A) The custom iQ5 shipping incubator loaded with the centrifugal system; (B) Schematics of the centrifugal system: (C) a photo of the Spin Disk Assembly (SDA); (D) a photo of the Motor Assembly (MA); (E) a photo of a "Smart" Cap inserted into a sample tube loaded to a tube holder; (F) a photo of a glass micropipette loaded with cyt-B reagent.

sample chambers, and each chamber has five tube holders to hold 2 ml sample tubes (Fig 2C). The chambers are sealed by screws and rubber gaskets as a secondary containment for shipping blood samples. Like the BA, the top of the SDA also has a hexagonal recess structure to allow insertion of the matching hexagonal structure at the bottom of the MA. There is also a magnet on the top surface of the SDA for rotation speed sensing. Finally, the MA plate rests on four brackets mounted on the wall of the incubator chamber and fixed by screws (Fig 2B). The MA has a DC motor in the center to spin the SDA through the bottom hexagonal structure (Fig 2D). A Hall effect magnetic sensor is used to measure the spin speed by sensing the magnet on the SDA, which also serves as a feedback to set motor power to reach the desired spin speed. A fan is run for 30 sec every 5 min to obtain a uniform temperature in the chamber. A custom-made printed circuit board (PBC) microcontroller is used to self-control the operation of the centrifugal system during transportation, i.e. the sample tubes will be spun for 60 sec at the 24-hour time point after starting the system and then left to stop afterwards; the target spin speed is usually reached within 15 sec. The maximum rotation speed of the system is 1500 rpm, beyond which the system becomes unstable. The whole system is powered by a 12 V NiMH battery pack.

The last component of the SSI is a "smart" cap for the 2 ml sample tube (Fig 2E). The "smart" cap is molded by polydimethylsiloxane (PDMS) with two holes to allow feedthroughs of a glass micropipette at the center and a venting needle on the side. The venting needle is necessary to balance the pressures inside and outside the tube to avoid disturbing the reagent in the micropipette. The tips of the micropipette and the needle sit at the center of the tube to minimize sample contact even when the tubes are tilted or inverted. Insertion of the cap into the sample tube will simultaneously seal the tube and two feedthroughs. The cyt-B reagent can be loaded into the micropipette (Fig 2F) from its base opening using a traditional pipette and released into the sample by centrifugal force.

## Cyt-B reagent release characterization and micropipette selection

Our automatic reagent release during transportation is implemented by a timed centrifugal spinning where the orifice of the micropipette serves as a microfluidic valve to release the reagent when a relative centrifugal force (RCF) threshold is passed. The valve should also be able to prevent premature release due to mechanical shocks during transportation. Burst valves and hydrophobic valves have been reported in centrifugal microfluidics [16], but they require microfabrication processes. On the other hand, micropipette with different orifice sizes are also commercially available. To simplify the process for our concept demonstration, commercial glass micropipettes (Fig 2F) were used in this project. Two orifice sizes (inner diameters 20 and 40 μm) and a fluorocoating on the 40 μm orifice were tested.

**Theoretical analysis.** Fig 3 shows the schematics of the micropipette spinning parameters. The reagent burst condition requires that the centrifugal pressure exceeds the burst capillary pressure. The centrifugal pressure $P_\omega$ can be expressed as:

$$P_\omega = \rho(RCF)H \cos \alpha = \rho \bar{r} \omega^2 H \cos \alpha \qquad (1)$$

where ρ is the density of the reagent, $\bar{r}$ is the average distance of the reagent to the rotation axis, ω is the rotation speed, H is the reagent column height inside the micropipette, and α is the tilting angle of the micropipette. The burst capillary pressure $P_c$ can be expressed as the sum of the burst capillary pressure at the orifice $P_{c,o}$ and the capillary pressure inside the micropipetter $P_{c,mp}$. There are two ways for the reagent to burst at the orifice [17]. $P_{c,o}$ is determined either by the outer diameter (OD) of the orifice $OD_o$ when the reagent-micropipette contact angle θ is small, or by the inner diameter (ID) of the orifice $ID_o$ when θ is large [17].

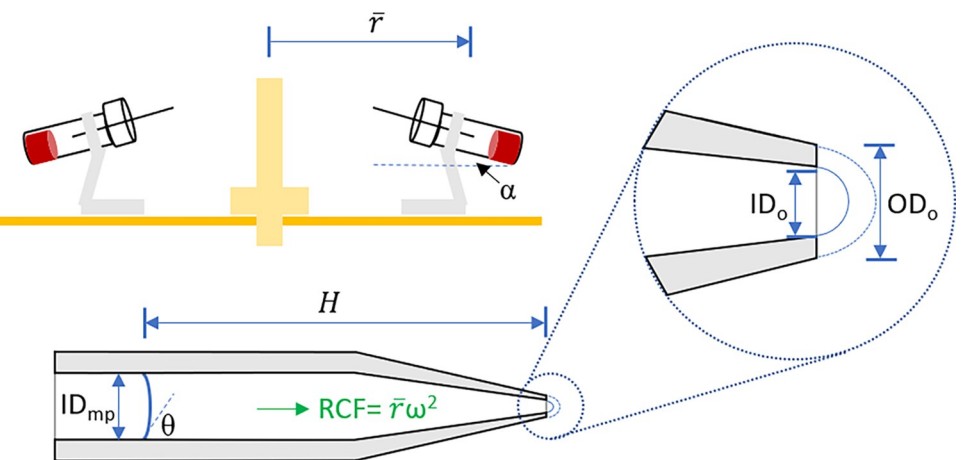

**Fig 3. Schematics of the micropipette parameters for cyt-B centrifugal release.**

As a result, $P_c$ can be expressed as:

$$P_c = P_{c,o} + P_{c,mp} = MAX\left\{\frac{4\sigma}{OD_o}, \frac{4\sigma \sin \theta}{ID_o}\right\} + \frac{4\sigma \cos \theta}{ID_{mp}} \tag{2}$$

where σ is the surface tension of the reagent and $ID_{mp}$ is the inner diameter of the micropipette (Note: the α tilting angle effects on the capillary pressures have been neglected). A characteristic crossover contact angle $\theta_c$ can be derived from Eq (2) at which the capillary pressures from the inner and outer diameters of the orifice are the same:

$$\theta_c = \sin^{-1} \frac{ID_o}{OD_o} \tag{3}$$

Finally, the overall reagent release will happen when $P_\omega > P_c$, i.e.:

$$\rho(RCF)H \cos \alpha = \rho \bar{r} \omega^2 H \cos \alpha > MAX\left\{\frac{4\sigma \sin \theta}{ID_o}, \frac{4\sigma}{OD_o}\right\} + \frac{4\sigma \cos \theta}{ID_{mp}} \tag{4}$$

**Experimental characterization.** To select the right micropipette orifice size, we experimentally characterized the cyt-B release through the glass micropipettes using a commercial centrifuge (Beckman Coulter Centrifuge 21R) so that higher RCF than our custom centrifugal system can also be tested. Dummy sample tubes with glass micropipette filled with 3.6 μL of cyt-B in Dimethyl Sulfoxide (DMSO) solution (the same reagent as that for CBMN assay) was loaded into a 15 ml centrifuge tube and spun for 60 sec at different RCFs. The height of the residual reagent was recorded, and the volume of the residual reagent was deduced from a calibration curve established for each micropipette based on the IDs of the micropipette at different distances away from the orifice.

The black and brown curves in Fig 4 indicate how the residual reagent volumes change with RCF for the purchased 20 μm and 40 μm orifice glass micropipettes. They show that there exists a RCF threshold for initial reagent release, as predicted by Eq (4). However, they also show that further release of the reagent requires continued increase of RCF. This can also be understood through Eq (4) because for our application the reagent height H in the micropipette will keep decreasing as the reagent is released off the micropipette. This is different from previously reported microfluidic chip capillary valves [16], where the reagent remains inside the device after passing the valves, and the reagent height can be maintained.

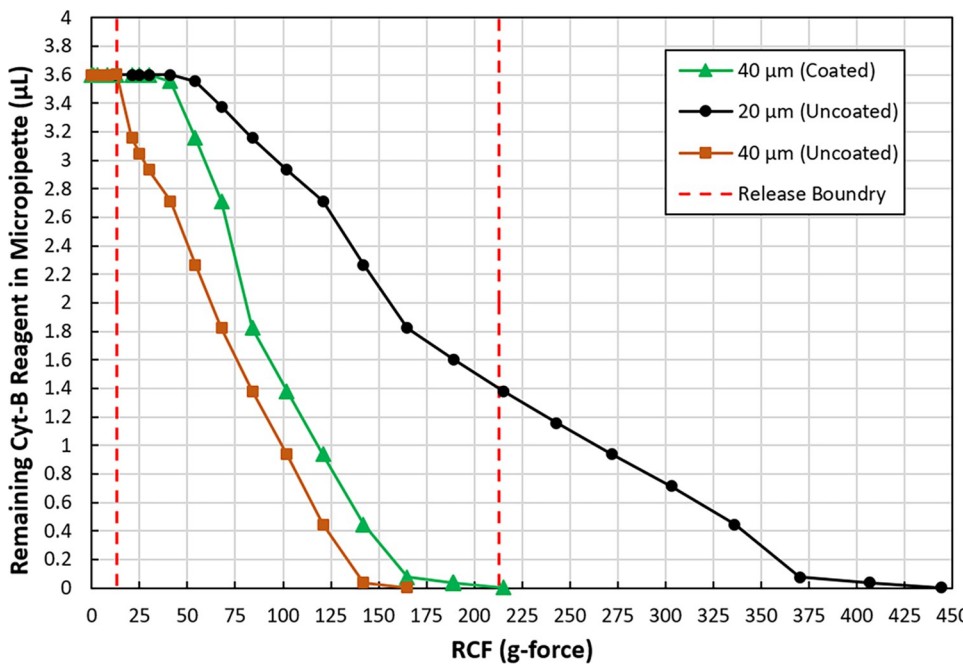

**Fig 4. Experimental characterization of cyt-B release at different RCF for micropipettes with different orifices (20 and 40 μm) and with a fluorocoating (40 μm).**

The two red dashed lines in Fig 4 indicate two critical parameters for cyt-B release. The lower value of $50^*\sin 15^o = 13$ g comes from the mechanical shocks the sample could experience during transportation ($> 50$ g, see next section) and the micropipette tilt angle α (15˚ for our system). The minimum burst RCF is expected to be higher than this to prevent premature cyt-B release. The higher value of 213 g comes from the highest RCF our centrifugal system can provide ($\bar{r}$ 8.75 cm and the maximum rotation speed of 1500 rpm give an RCF of 220 g) and the micropipette tilt angle α ($220^*\cos 15^o = 213$ from Eq (1)). We can see that the 40 μm orifice micropipette (brown curve) with a minimum burst RCF of 13 g is just on the edge to cause premature cyt-B release, and the 20 μm orifice micropipette could not release all the cyt-B at the highest RCF of 213 g.

To address these issues, we further tested a 40 μm orifice with a fluorocoating, as shown by the green curve in Fig 4. The coated micropipette showed an initial burst RCF of ~ 41 g (high enough to prevent accidental cyt-B release) and >99% release of the reagent at 213 g RCF. To understand the effect of the fluorocoating, the contact angles of the cyt-B reagent to a bare glass and a glass with a fluorocoating were measured to be 37˚ and 68.6˚. According to the micropipette manufacturer, the pipette orifice has an $OD_o/ID_o$ ratio of 1.3, which gives a $θ_c$ value of 50˚ by Eq (3). That means for the uncoated micropipettes, the capillary pressures are determined by the outer diameters of the orifices, which are 26 μm and 52 μm respectively. With a fluorocoating, the capillary pressure is determined by the $ID_o$, which is 40 μm for the 40 μm orifice, or an equivalent of $40/\sin 68.6^o = 43$ μm if the contact angle is considered. This value lies between the previous two diameter values and has shown to satisfy both the low and high RCF requirements. Therefore, the 40 μm orifice micropipette with a fluorocoating was selected for our SSI application.

## Resistance to mechanical shocks against premature cyt-B release

To prevent premature cyt-B release, we first characterized possible mechanical shocks the sample tubes could experience during transportation. Considering the weight and size of the iQ5

shipping incubator, we expect the incubator to be handled through the two hand pockets on the external left and right sidewalls. That means that the SSI is unlikely to be inverted, and we also assume that it might not be dropped from a height greater than 1 ft. With these assumptions, we simulated possible mechanical shocks by dropping the SSI vertically at different heights (4, 6, 8, 12 inches), as well as a sideway tilt fall. The mechanical shocks along vertical direction and radial direction were recorded at different locations of the SDA using 25G and 50G Drop-N-Tell shock indicators (Teletemp Corp., Part# TD25R and TD50R), as shown in Fig 5A. The maximum g-forces of the triggered shock indicators out of three trials for different conditions are shown in Fig 5B. The shock indicators were mainly triggered along the vertical z-direction and the shock can exceed 50 g depending on the drop height. No indicator was triggered along radial direction and for the sideway tilt fall.

To verify that the mechanical shocks would not cause premature cyt-B release, a 40 μm orifice micropipette with fluorocoating loaded with cyt-B reagent and mounted on a dummy sample tube inside the SDA also went through the vertical drops and sideway tilt fall. Volumes of the reagent were measured by the heights of the reagent inside the micropipettes using a ruler. Fig 5C shows how the volume changed with consecutive mechanical shocks. The error

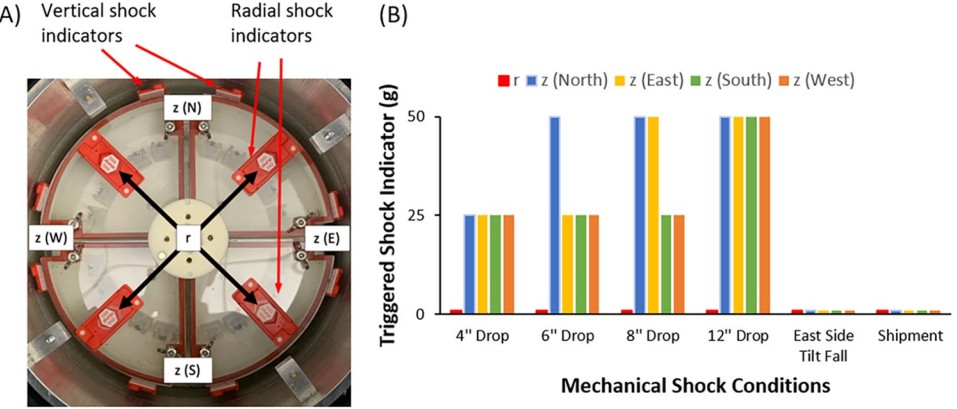

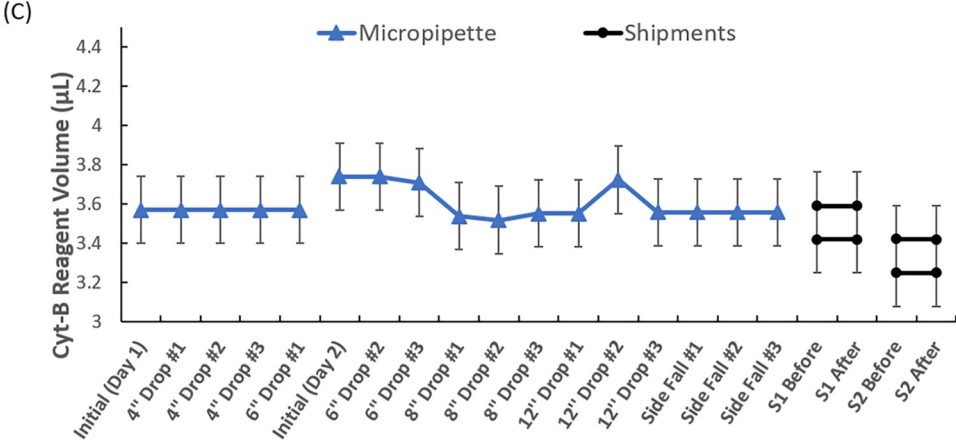

**Fig 5. Assessing the ability of the 40 μm micropipette with fluorocoating against premature reagent release under mechanical shocks.** (A) top view of the SDA with mounted 25g and 50g shock indictors at different locations. (B) Maximum g-force of the shock indicator triggered for each condition from three trials. (C) Reagent volumes inside the micropipettes under different conditions demonstrate the resistance to premature reagent release by the micropipette valve.

bar in the plot is ±0.17 $\mu l$, constant for all data points and calculated by our estimated height measurement error (±0.5 $mm$) and an $ID_{mp}$ of 0.66 mm. The reagent volume of the micropipette did not change significantly comparing with its initial value before the mechanical shocks.

Finally, we verified the micropipette's resistance to mechanical shock through two real shipments (two micropipettes per shipment) before performing CBMN assays. The SSI box was shipped from The University of Arizona's College of Medicine -Phoenix campus (Latitude: 33.453108; Longitude: -112.067381) to HonorHealth Research and Innovation Institute (Latitude: 33.581396; Longitude:-111.883099) at Scottsdale, AZ (~ 20 miles away), and then shipped back the following day using FedEx SameDay City Standard services. The SSI was programmed without centrifugation during the shipments. Fig 5B shows that the mechanical shocks during the FedEx SameDay City shipments were much smaller than we expected, and none of the shock indictors were triggered. Fig 5C also shows that the volumes of the reagents were not changed after the shipments, indicating that the micropipette valve works well to prevent premature reagent release.

## pH effect on the number of binucleated cells

Obtaining enough binucleated-cells (BN) is an important part of developing a valid dose curve for CBMN assay. For an accurate micronuclei per binucleated cell ratio (MN/BN) for a given dose, at least 500 BN cells or 100 MN should be counted [18]. Our preliminary counts of BN cells in the SSI following a 54-hr cell culture protocol [14] were less than 500 despite optimization of our harvesting and imaging protocols. On the other hand, more than 500 BN cells were counted for the control samples cultured inside a traditional $CO_2$ incubator.

The main function of the $CO_2$ gas for cell culture is to maintain the pH of the culture medium through the bicarbonate buffering system. pH has been reported to have a significant effect on lymphocyte proliferation [19]. Although it is possible to culture cells without $CO_2$ for biodosimetry assays [9, 20], the sample tube has to be tightly closed, which is not the case for our setup since our sample tube contains a venting needle. To address the issue of culture without $CO_2$ and confirm the effect of pH, 25 mM HEPES buffer was used to maintain the pH during cell culture in the SSI [21] and compared to the cell cultures in the SSI without HEPES and in a laboratory $CO_2$ incubator. 100 μl fingerstick blood in 500 μl PB-MAX medium was used for each sample tube. Three sample tubes were processed for each culture condition. Fig 6 shows the numbers of BN cells for the three culture conditions. The result shows that addition of the HEPES buffer did significantly increase the number of BN cells for SSI cell culture (p = 0.0038) with a value well above 500. The number is still lower than that of the traditional $CO_2$ incubator, but is not statistically significant (p = 0.0609). With these results, cell culture with HEPES buffer was used for the SSI CMBN assay.

## SSI demonstration of reagent handling during transportation for CBMN assay

With enough BN cells, we went further to demonstrate automatic reagent handling during transportation by SSI for the "in-transit" CBMN biodosimetry logistics. Fingerstick blood was collected and irradiated by an X-ray irradiator with doses of 0, 1, 2, 3 and 4 Gy. The sample was split into two groups. One group was cultured inside the SSI with cyt-B release at 24-hour time point by centrifugation during transportation. The SSI was shipped from The University of Arizona College of Medicine–Phoenix campus to HonorHealth Research and Innovation Institute (Scottsdale, AZ) and then shipped back the next day, both by FedEx SameDay City Standard services as we did before. The other group was cultured using a conventional $CO_2$

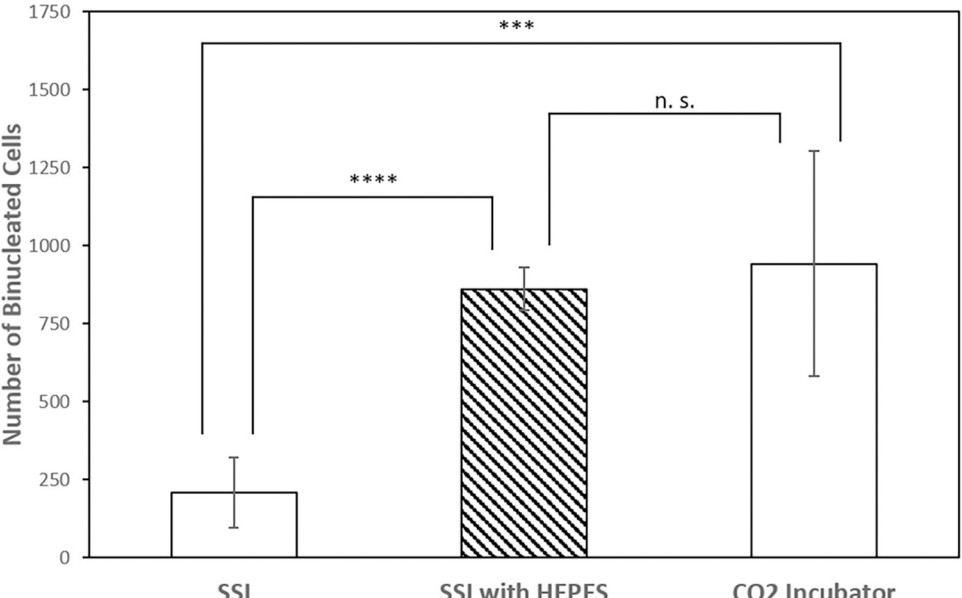

**Fig 6. The number of BN cells under three culture conditions: In SSI with and without HEPES buffer, and in a traditional $CO_2$ incubator.** *** represents a p-value less than 0.001 and **** represents a p-value less than 0.0001. "n. s." stands for "not significant" (p-value larger than 0.0167).

incubator with cyt-B added manually at 24-hour time point. All the samples were harvested after 54 hours of culture and then analyzed for CBMN assay. Three shipments were conducted with each shipment containing blood samples from one donor (total three donors). The average of the three shipments was used to plot the dose curve. Fig 7A shows the dose curves for SSI and the conventional process. Clear dose-dependent MN/BN ratios have been observed both by SSI and the conventional control. Fig 7B shows that there is a strong correlation between the MN/BN ratios obtained using the SSI and a control process with a Pearson's correlation coefficient of P = 0.9972 and 95% confidence interval of (0.9540, 0.9998). This suggests

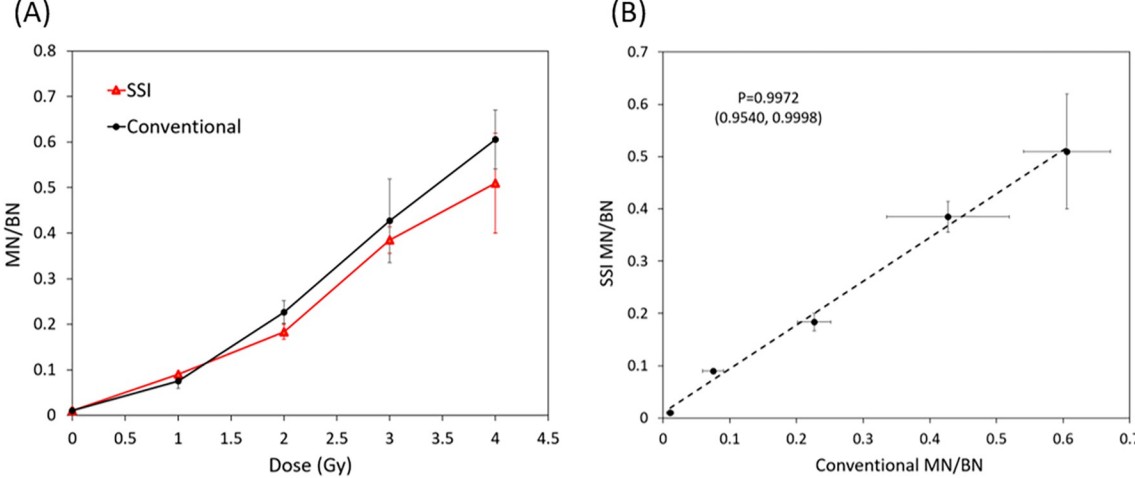

**Fig 7.** (A) CBMN dose curves by SSI and conventional process showing strong dose-dependent MN/BN ratios. (B) Pearson's correlation analysis showing a strong positive correlation between the SSI and the conventional control samples with a coefficient P = 0.9972 and 95% confidence interval (0.9540, 0.9998).

that the SSI with automatic cyt-B release during transportation could be used for CBMN assay for biodosimetry triage.

## Discussion

In this project, we demonstrated the feasibility of active reagent handling and sample pre-processing during specimen transportation using a centrifugal microfluidic system inside a temperature-controlled environment. The technology was used to ensure a reliable CBMN biodosimetry assay that allows immediate sample culture right after collection to dramatically shorten the assay response time, which could have a significant impact on rapid radiation triage after a large scale nuclear/radiological event. The technology should also be applicable to the gold-standard DCA cytogenetic dosimetry assay due to its similar cell culture and reagent (Colcemid) addition requirements [9].

Several improvements of the current system are foreseen. First, even though the estimated the cost of the centrifugation system (including motor, battery pack, microcontroller board, mechanical parts etc.) is moderate (< $150), the currently used iQ5 shipping incubator with a rechargeable NiMH battery pack can be heavy and expensive (~ $5k). For national stockpiling, cheaper and disposable long shelf life alkaline battery will be advantageous over the NiMH rechargeable battery. (Please note that both batteries types are compatible with transportation, including air, which is different from lithium batteries that are regulated [22].) Recently, we reported a shipping incubator optimized for national stockpile based on traditional vacuum-insulated-panel shipping boxes and alkaline battery resistive heating with a much reduced cost of < $400 [15]. We expect a future SSI to be implemented using the newly developed transportation container.

Secondly, the current sample holder is still bulky with an SSI capacity of ~ 20 samples. For mass incident scenario, it is important to scale up the number of samples per shipment for the real-world application. There are several aspects in terms of scaling up the sample size per shipment. For the centrifugal system, it has the advantage that it is easy to scale up as no additional pumps are required. For the sample tubes, we do expect miniaturized sample tubes in a high-throughput rack format (e.g. the 96-tube Matrix rack used in literature [2, 15]) to be used in the future. For the micropipettes, the current ones were made by low throughput glass capillary pulling, which costed ~ $10/each. We expect future ones to be manufactured through high throughput microfabrication [23] to bring the size and cost down.

Third, in the current demonstration, only inter-city ground shipments were tested. In a previous study, we did perform mechanical testing of a low-cost shipping incubator for long-distance air shipments [15]. It is interesting to note that the mechanical shocks from the FedEx SameDay City Standard services (< 25 g) were much smaller than the mechanical shocks (up to 100 g) we observed previously during flight shipments by FedEx Express services [15]. This could come from careful handling of the shipping container by FedEx SameDay City Standard services. However, even for 100 g vertical impact (26 g along the micropipette), the 40 μm fluorocoated micropipette should still be able to hold the cyt-B reagent. Long term titling and inversion of the shipping container may impose a challenge, but considering the fact that the container holds large number of human samples from a mass nuclear/radiation event, it is expected that special handling would be provided during emergency response and logistics to address this issue, which could also minimize the mechanical shocks at the same time.

In summary, we expect the SSIs be stockpiled at all major governmental emergency response sites across the country to be rapidly deployed to a nuclear incident site for collecting biospecimen and shipping them to centralized cytogenetic laboratories for high-throughput biodosimetry triage for guiding radiation countermeasures during a nuclear/radiation disaster.

With further development, the future SSI using the low-cost shipping incubator with miniaturized sample tubes and micropipettes could accommodate multiple (e.g. > 10) 96-tube sample racks with a cost of ~ $100-200/rack, which corresponds to 1–2 million dollars per 1 million people. Such estimated cost is within the budget of the U.S. state/federal government [15].

Finally, besides radiation biodosimetry, transportation is needed in many biomedical applications [24]. The concept of actively processing sample during transportation adds a new feature in providing high quality samples in a time-sensitive manner with potential economic benefit (e.g. reduction in cold chain requirements) and may find applications in other healthcare scenarios.

# Materials and methods

## Ethics statement

All blood collections were approved by the institutional review boards at The University of Arizona (IRB# 1708743060) and performed after written informed consents were given. All experiments were performed in accordance with the relevant guidelines and regulations.

## Custom centrifugal system

The custom centrifugal system was made in-house. The ball bearing of the BA was purchased from McMaster-Carr (Cat# 6383K53). The DC motor was a 540 Crawler Brushed Motor 80T from RC4WD (Visalia, CA; Part# Z-E0001). The fan was a 12V fan from Sunon Fans (Product# ME40101VX-000U-A99). The 12V battery pack consisted of eight SANHR3U AA NiMH rechargeable 2500 mAH cells, purchased and assembled at Batteries Plus LLC. The Hall effect magnetic sensor was purchased from Digi-Key (Honeywell Sensing and Productivity Solutions, Product# SM351RT). The PCB of the microcontroller was machined by a LPKF ProtoMat S62 circuit plotter. The microcontroller was assembled in-house. The schematics of the circuits and 3D print of the microcontroller can be found in S1 Fig. All other plastic parts were fabricated in-house either by computer numerical control (CNC) machining or by 3D printing.

## "Smart" cap fabrication

The "smart" cap was made by molding PDMS (Dow Sylgard 184 with 10:1 mix ratio) from an air pocket formed by three plastic plates. Two dummy needles were used to mold the feedthroughs. After the PDMS cap was fully cured, a working glass micropipette and a 25-gauge venting needle (BSTEAN blunt tip needle) were inserted through the feedthroughs to finish the "smart" cap fabrication.

## Micropipette and cyt-B reagent

The 20- and 40-μm ID glass micropipettes were purchased from FivePhoton Biochemicals (San Diego, CA) with a base OD of 1 mm, base wall thickness of 0.17 mm, length of 4.5 cm, and OD:ID ratio of 1.3 at tip. Cyt-B reagent was made by diluting a 10 mg/ml stock solution of cyt-B in DMSO (Sigma Aldrich, Cat# C2743) to a concentration of 1 mg/ml using DMSO purchased from ATCC (Part# 4-X-5). The sample tubes were 2 ml microcentrifuge tubes from VWR (Cat# 20170–170). To load Cyt-B reagent into the glass micropipette, the reagent was manually pipetted into the micropipette using a 10 μl plastic pipette tip from the base of the glass micropipette.

## Micropipette fluorocoating and contact angle measurement

To prepare fluorocoating surface for the glass micropipettes, the micropipettes were treated with $O_2$ plasma using an Oxford Plasmalab 80 Plus reactive ion etcher with gas flow 50 sccm, pressure 500 mtorr, power 50 W for 2 minutes. Then the micropipettes were transferred to a container on top of a 45°C hotplate, and ~ 200 μl of (tridecafluoro-1,1,2,2-tetrahydrooctyl) trichlorosilane (Gelest, Inc, Cat# SIT8174.0) was dispensed inside the container to vapor coat the micropipette surface for 30 minutes. The contact angles of the cyt-B reagent on bare and fluorocoated glass slides were measured by a ramé-hart instrument co. (Succasunna, NJ) goniometer and the DROPimage software.

## BN cell number optimization

To optimize the number of BN cells for CBMN assay, 100 μl fingerstick blood was collected into a heparinized Minivette capillary tube (SARSTEDT AG & Co, Part# 17.2112.101), and then dispensed into the 2 ml sample tube containing 500 μl of PB-MAX medium (Thermo-Fisher Scientific, Cat# 12557013). For $CO_2$ control culture, three sample tubes were loosely capped and cultured inside a Forma Series 3 water jacketed incubator at 37°C with 5% $CO_2$ (ThermoFisher Scientific). For SSI culture, three sample tubes were also loosely cap (no 'smart' cap was used) and loaded into the SSI sample tube holders inside the SDA compartments to be cultured at 37°C. For another three sample tubes, 15 μl of 1M HEPES buffer (Sigma-Aldrich, Cat# 83264) was added to each tube to reach a final concentration of 25 mM and cultured in a similar fashion inside SSI. Paper towels wetted by 5 ml of DI water were used inside each compartment to prevent drying of the sample. The compartments were sealed by rubber gaskets and screws. 3.6 μl of 1 mg/ml cyt-B reagent was added to each sample tube manually to reach a final concentration of 6 μg/ml concentration at 24-hour time point for all conditions. Samples were then harvested after another 30-hour culture and analyzed according to the harvesting and imaging protocols. All the sample tubes were autoclaved before use.

## CBMN assay by SSI

For each shipment, fingerstick blood from one donor was collected into ten 100 μl Minivette capillaries as mentioned before. The blood samples inside the capillaries were irradiated at 0, 1, 2, 3, 4 Gy (two capillaries for each dose) with a dose rate of 0.7 Gy/min by an X-Rad320 irradiator (Precision X-Ray, Madison, CT) with an acceleration voltage of 320 kV, 8 mA current, 43 cm shelf source distance and a Al/Cu/Sn filter. The samples were then dispensed into the sample tubes containing appropriate culture medium and cultured half in SSI and half in the conventional $CO_2$ incubator at 37°C. Cyt-B was added to the sample during transportation automatically by the "smart" cap and the centrifugal system in SSI (confirmed by the empty micropipettes after shipments), and manually for the conventional process at 24-hour time point. All the samples were harvested after 54 hour culture and imaged for the CBMN assay. Total three shipments were conducted. The three donors contain two males and one female at the ages of 21 to 47.

## Lymphocyte harvesting

After cell culture, the sample tubes were housed inside 15 ml tubes and centrifuged at 240 g for 5 min. 500μL of supernatant was removed from each tube. 800μL of 0.075M KCl hypotonic solution was then added to each tube and incubated at room temperature for 10 min to lyse the red blood cells. Afterwards, 200 μl methanol:acetic acid fixative solution (3:1 ratio) was added to each sample and pipetted 10 times. Each sample was then divided into three aliquots

of 333μL and placed into a 96-well plate to avoid lymphocyte aggregation in the subsequent fixation/washing steps. The fixation/washing steps were done by centrifuging the plate at 150 g for 1 min, followed by the replacement of 50% of the supernatant with new fixative and cell resuspension. The process was repeated five times or until the solution appeared colorless.

### Imaging and image analysis

Each aliquot of 333 μl was transferred to a 600 μl centrifuge tube and the cells were spun down at 63 g for 5 min. Supernatant was removed with ~ 10 μl liquid remaining. Cells were resuspended by vortexing, and the 10 μl cell suspension was dropped onto a clean glass slide to spread the cells. The slide was mounted with a cover glass and mounting medium (VectaShield HardSet H-1500 from Vector Laboratories). The mounting medium contains DAPI, which stains the nuclei of the cells. The cover glass was sealed by a coverslip sealant from Biotium (Cat# 23005). For extended storage, the slides were kept in a 4°C fridge.

The slides were imaged by a Zeiss AXIO Imager M2 microscope using a 20x objective, a DAPI filter set and the tiling feature of Zen software. Total 15 mm by 15 mm area of the slide was scanned to cover the entire sample area. BN cells and their respective MN were manually counted using the Zen software. They were identified based on the constraints reported in literature [25], i.e. a) the two nuclei are close to each other, similar in shape and size, and not overlapping, b) the micronuclei are relatively circular in shape, less than one nucleus diameter away from the nuclei, and each micronucleus is less than 20% the size of one of the two nuclei. A typical BN cell with micronuclei can be seen in S2 Fig. For purposes of BN cell optimization in the SSI, BN cells were counted for all three aliquots/slides. But for the purposes of producing a dose curve, only the first 500 BN cells or the first 100 MN were counted.

For the BN cell optimization experiment, an ANOVA test was conducted using Microsoft Excel for the 3 samples for each condition to see if there was a statistically significance difference present in the group. This was followed by a Bonferroni post Hoc T-test to assess the difference between each of the three condition comparisons. For the SSI and conventional CBMN assays, the Pearson's correlation coefficient was calculated by Excel and the confidence interval were calculated by R software.

## Supporting information

**S1 Fig. Schematics of the PBC microcontroller circuits.** (A) a power reduction circuit from 12 to 5 V to provide power to the MCU (microcontroller unit) chip; (B) the clock circuit for timekeeping; (C) the MCU pinout and circuit diagram; (D) the fan control circuit; with an additional diode (red), the same circuit design was also used to control the motor; (E) 3D print of the PCB microcontroller (top).
(TIF)

**S2 Fig. A typical image of a binucleated cell with micronuclei.**
(TIF)

## Acknowledgments

The authors would like to thank Baiju Thomas at The University of Arizona for helping with plastic part fabrication, Jamie Cox at HonorHealth Research and Innovation Institute for receiving and shipping the SSI, and the NIH/NIAID Center for Medical Countermeasures Against Radiation (CMCR) program (Project Number 5U19AI067773) for CBMN assay protocol.

## Author Contributions

**Conceptualization:** Jian Gu, Adarsh Ramakumar, Frederic Zenhausern.

**Formal analysis:** Adam R. Akkad, Jian Gu, David J. Brenner.

**Funding acquisition:** Jian Gu, Adarsh Ramakumar, Frederic Zenhausern.

**Investigation:** Adam R. Akkad, Jian Gu, Brett Duane, Alan Norquist.

**Methodology:** Jian Gu.

**Project administration:** Jian Gu, Frederic Zenhausern.

**Resources:** Frederic Zenhausern.

**Software:** Brett Duane.

**Supervision:** Jian Gu.

**Validation:** Adam R. Akkad.

**Visualization:** Adam R. Akkad, Jian Gu.

**Writing – original draft:** Adam R. Akkad.

**Writing – review & editing:** Jian Gu, Brett Duane, Alan Norquist, David J. Brenner, Adarsh Ramakumar, Frederic Zenhausern.

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
