## [Decision Letter · Decision Letter 0]

11 Apr 2022

PONE-D-22-08129Automatic Reagent Handling and Assay Processing of Human Biospecimens inside a Transportation Container for a Medical Disaster Response Against RadiationPLOS ONE

Dear Dr. Gu,

Thank you for submitting your manuscript to PLOS ONE. After careful consideration, we feel that it has merit but does not fully meet PLOS ONE’s publication criteria as it currently stands. Therefore, we invite you to submit a revised version of the manuscript that addresses the points raised during the review process.

We look forward to receiving your revised manuscript.

Kind regards,

Talib Al-Ameri, Ph.D

Academic Editor

PLOS ONE

Journal Requirements:

2. We note that you have a patent relating to material pertinent to this article. Please provide an amended statement of Competing Interests to declare this patent (with details including name and number), along with any other relevant declarations relating to employment, consultancy, patents, products in development or modified products etc. Please confirm that this does not alter your adherence to all PLOS ONE policies on sharing data and materials, as detailed online in our guide for authors http://journals.plos.org/plosone/s/competing-interests by including the following statement: "This does not alter our adherence to  PLOS ONE policies on sharing data and materials.” If there are restrictions on sharing of data and/or materials, please state these. Please note that we cannot proceed with consideration of your article until this information has been declared.

"for the financial support from the

439 Congressionally Directed Medical Research Programs, Merit Based Discovery Award (Award# W81XWH-15-2-0076, Grant PR142006)."

"The funding was awarded to J.G. and F.Z. by a subaward (Agreement No.: 3042) from Armed Forces Radiobiology Research Institute through a Congressionally Directed Medical Research Program, Merit Based Discovery Award to A.R. (Award# W81XWH-15-2-0076, Grant PR142006). The funders (other than the named authors) had no role in study design, data collection and analysis, decision to publish, or preparation of the manuscript."

Reviewers' comments:

Reviewer's Responses to Questions

**Comments to the Author**

1. Is the manuscript technically sound, and do the data support the conclusions?

Reviewer #1: Yes

2. Has the statistical analysis been performed appropriately and rigorously? 

Reviewer #1: Yes

3. Have the authors made all data underlying the findings in their manuscript fully available?

Reviewer #1: Yes

4. Is the manuscript presented in an intelligible fashion and written in standard English?

Reviewer #1: Yes

5. Review Comments to the Author

Reviewer #1: This manuscript describes the development of a shipping container that would allow blood cell culturing during transportation from the site of collection to the laboratory where processing would occur. This type of device would shorten the time to dose estimates for biodosimetry assays and would have an impact on large scale events. The concept of the device is very interesting, however, there are a few unanswered questions below:

1. These devices seem large, heavy and potentially costly. There was not indication of the cost of this box.

2. The authors stated that more tubes could be loaded in each box. This would be essential for mass casualty situations. It would be nice to know an estimate of the tube capacity of the system.

3. The shipping scenario tested was ground based and quite short. Has the container been tested on flights or ground transportation of longer duration? It has been tested for drops and tilts but what about extreme vibrations that might be experienced on flight take off/landing?

4. Fig 2, F – this image is hard to make out. Overall the resolution of the figures is poor.

5. Are there any issues with shipping devices by plane that contain batteries? Has the device been discussed with transportation authorities

6. Is there a vision for how these devices would be deployed? For example, would they be stockpiled in one location and sent to the site of the incident?

7. Since the MN/BN at the high end of the dose response curve is lower for the SSI than for the conventional method, would you recommend having a calibration curve generated for this type of cell processing?

6. PLOS authors have the option to publish the peer review history of their article (what does this mean?). If published, this will include your full peer review and any attached files.

Reviewer #1: No

---

## [Author Response · Author response to Decision Letter 0]

26 Apr 2022

April 22, 2022

RE: Rebuttal letter for manuscript PONE-D-22-08129 “Automatic Reagent Handling and Assay Processing of Human Biospecimens inside a Transportation Container for a Medical Disaster Response Against Radiation”

Dear PLOS ONE Editor:

We have revised our manuscript according to the Editor and reviewer’s comments. The changes and comments are summarized below:

1. Ensure manuscript meets PLOS ONE’s style requirements

a. The title has been revised according PLOS ONE format guidelines;

b. Corresponding authors have been revised according guidelines;

c. The headings have been revised with three tiers, font sizes, sentence style etc. according to guidelines;

d. Figure citations and captions have been revised according to guidelines;

e. In text reference citations have been revised to brackets according to guidelines; they were also reformatted in the “References” section;

f. Supporting information materials have been named according to guidelines;

2. Amended statement of Competing Interests to declare patent

The new declaration statement can be:

I have read the journal’s policy and the authors of this manuscript have the following competing interests: F.Z., B.D., J.G., A.N., D.J.B. have filed a patent application PCT/US2019/64737, titled “Smart Storage container for Health Logistics”. This does not alter our adherence to PLOS ONE policies on sharing data and materials.

3. Funding info in Acknowledgements

The funding information has been removed from the “Acknowledgements”. The current Funding Statement is ok.

4. Review reference list

The references are all correct. They have been reformatted using a better software. One new reference was added regarding regulation on lithium battery (Ref [22]).

Reviewer’s comments to the Author (Reviewer #1):

1. These devices seem large, heavy and potentially costly. There was not indication of the cost of this box.

We added the comments of cost in “Discussion” line 321-358. The container could be large like now due to the fact of accommodating large number of samples. The weight and cost are some limitations of the current iQ5 shipping incubator, which should be replaced in the future by a new shipping container we reported recently. Future development aims to reduce the overall cost to ~ $100-200/96 sample rack.

2. The authors stated that more tubes could be loaded in each box. This would be essential for mass casualty situations. It would be nice to know an estimate of the tube capacity of the system.

More tube capacity per SSI and cost depend on future miniaturization of the tube and micropipette, which we explained more in “Discussion” line 331-339. We expect >10x 96-tube racks could be accommodated in the future.

3. The shipping scenario tested was ground based and quite short. Has the container been tested on flights or ground transportation of longer duration? It has been tested for drops and tilts but what about extreme vibrations that might be experienced on flight take off/landing?

We elaborated more about this limitation in “Discussion” line 340-352. We haven’t tested current container for longer flight transportations, which should be done in further study. However, the mechanical shocks collected from a previous flight test of another container from a round-trip New York, NY to Phoenix, AZ and back should include the “extreme vibrations that might be experienced on flight take off/landing” the reviewer mentioned, which had a value that should be tolerated by this container. We also expect special handling to be provided to these containers (containing large amount of biospecimen) during a nuclear event that could provide decreased mechanical shocks if needed.

4. Fig 2, F – this image is hard to make out. Overall, the resolution of the figures is poor.

We have saved the image with a higher resolution. A larger area of the micropipette with cyt-B was also included so that the reader/reviewer can make sense of the image better. We also increased the resolution/size of Fig 3-7 for better viewing purpose.

5. Are there any issues with shipping devices by plane that contain batteries? Has the device been discussed with transportation authorities?

We added in “Discussion” line 325-327 that NiMH battery used in this project is not actively regulated (guideline from IATA, International Air Transport Association). We haven’t had discussion with transportation the authorities yet. In the context of our previous container development, we had discussions with experts at FedEx and from these internal discussions, it was assessed that our system was enclosed in multiple layers of containments with minimal risk to transportation, and to the best of our knowledge, was not in conflict with any regulation. We will further confirm this with the transportation authorities in future effort toward potential scale-up fabrication and commercialization.

6. Is there a vision for how these devices would be deployed? For example, would they be stockpiled in one location and sent to the site of the incident?

We added the vision of how to use the SSI in “Discussion” line 353-355. We expect SSIs be stockpiled at all major metropolitan areas around the U.S., deployed to the incident site from the closest locations and shipped to cytogenetic labs for analysis. We also had informal discussions with lab personnel at Sonoran Quest Laboratories (Phoenix AZ) who express interest using such of our containers for ground-based logistical applications in more conventional operations. If so, it is foreseeable to have also some of these devices available at distributed sites in clinical laboratories which could be leveraged in the case of an emergency response. 

7. Since the MN/BN at the high end of the dose response curve is lower for the SSI than for the conventional method, would you recommend having a calibration curve generated for this type of cell processing?

Yes, the current practice is that each cytogenetic laboratory generates its own calibration curve due to the differences of the assay protocol from each lab, such as reported in the “RENEB (Realizing the European Network of Biodosimetry) intercomparison exercises analyzing micronuclei” by Depuydt et al. (2017).

I hope we have addressed all the concerns raised during the review process. Should you have any other questions, please feel free to let me know by email: jgu10@arizona.edu or phone 602-827-5950.

Best regards,

Jian Gu, Ph.D.

Associate Professor

Center for Applied NanoBioscience and Medicine

Department of Basic Medical Sciences

The University of Arizona College of Medicine-Phoenix

475 N 5th St, BSPB/Rm E612, Phoenix, AZ 85004

Tel: (602) 827-5950; Fax: (602) 827-2122

---

## [Decision Letter · Decision Letter 1]

3 May 2022

Automatic reagent handling and assay processing of human biospecimens inside a transportation container for a medical disaster response against radiation

PONE-D-22-08129R1

Dear Dr. Gu,

We’re pleased to inform you that your manuscript has been judged scientifically suitable for publication and will be formally accepted for publication once it meets all outstanding technical requirements.

Kind regards,

Talib Al-Ameri, Ph.D

Academic Editor

PLOS ONE

Reviewers' comments:

Reviewer's Responses to Questions

**Comments to the Author**

1. If the authors have adequately addressed your comments raised in a previous round of review and you feel that this manuscript is now acceptable for publication, you may indicate that here to bypass the “Comments to the Author” section, enter your conflict of interest statement in the “Confidential to Editor” section, and submit your "Accept" recommendation.

Reviewer #1: All comments have been addressed

2. Is the manuscript technically sound, and do the data support the conclusions?

Reviewer #1: Yes

3. Has the statistical analysis been performed appropriately and rigorously? 

Reviewer #1: N/A

4. Have the authors made all data underlying the findings in their manuscript fully available?

Reviewer #1: Yes

5. Is the manuscript presented in an intelligible fashion and written in standard English?

Reviewer #1: Yes

6. Review Comments to the Author

Reviewer #1: The authors have addressed all of my comments. This paper can now be accepted for publication. Thank you.

7. PLOS authors have the option to publish the peer review history of their article (what does this mean?). If published, this will include your full peer review and any attached files.

Reviewer #1: No

---

## [Editor Report · Acceptance letter]

13 May 2022

PONE-D-22-08129R1 

Automatic reagent handling and assay processing of human biospecimens inside a transportation container for a medical disaster response against radiation 

Dear Dr. Gu:

I'm pleased to inform you that your manuscript has been deemed suitable for publication in PLOS ONE. Congratulations! Your manuscript is now with our production department. 

Kind regards, 

on behalf of

Dr. Talib Al-Ameri 

Academic Editor

PLOS ONE